# A Novel Recombinant Human Filaggrin Segment (rhFLA-10) Alleviated a Skin Lesion of Atopic Dermatitis

**DOI:** 10.3390/bioengineering11050426

**Published:** 2024-04-26

**Authors:** Jiawen Zhu, Xinhua Zhong, Hui Liao, Jianhang Cong, Qiqi Wu, Shuang Liang, Qi Xiang

**Affiliations:** 1State Key Laboratory of Bioactive Molecules and Drug Gability Assessment, Jinan University, Guangzhou 510632, China; 13560646034@163.com (J.Z.); zxh0126@stu2022.jnu.edu.cn (X.Z.); 18028636361@163.com (H.L.); 2022c@stu2022.jnu.edu.cn (J.C.); wuqiqi@stu2021.jnu.edu.cn (Q.W.); 2021103898ls@stu2021.jnu.edu.cn (S.L.); 2Institute of Biomedicine and Guangdong Provincial Key Laboratory of Bioengineering Medicine, Jinan University, Guangzhou 510632, China; 3Biopharmaceutical R&D Center, Jinan University, Guangzhou 510632, China

**Keywords:** atopic dermatitis, recombinant human filaggrin segment, P815, percutaneous penetration, skin repair

## Abstract

Atopic dermatitis (AD), a prevalent chronic inflammatory skin disorder, is marked by impaired skin barrier function and persistent pruritus. It significantly deteriorates patients’ quality of life, making it one of the most burdensome non-lethal skin disorders. Filaggrin plays a crucial role in the pathophysiology of barrier disruption in AD, interacting with inflammatory mediators. It is an integral part of the extracellular matrix architecture, serving to protect the skin barrier and attenuate the inflammatory cascade. In this study, we engineered a novel recombinant human filaggrin (rhFLA-10) expression vector, which was subsequently synthesized and purified. In vitro and ex vivo efficacy experiments were conducted for AD. rhFLA-10, at low concentrations (5 to 20 μg/mL), was non-toxic to HACaT cells, significantly inhibited the degranulation of P815 mast cells, and was readily absorbed by cells, thereby exerting a soothing therapeutic effect. Furthermore, rhFLA-10 demonstrated anti-inflammatory properties (*p* < 0.05). In vivo, efficacy experiments further substantiated that rhFLA-10 could effectively ameliorate AD in mice and facilitate the repair of damaged skin (*p* < 0.001). These findings underscore the considerable potential of rhFLA-10 in the treatment of AD.

## 1. Introduction

Atopic dermatitis (AD), a prevalent chronic inflammatory skin disease, is characterized by skin barrier dysfunction and severe, recurrent pruritus, significantly impairing patients’ quality of life. Often the precursor to other atopic diseases such as allergic rhinitis, coughing, and asthma, AD affects an estimated 230 million people globally, making it a leading cause of non-fatal disease burden among skin diseases [1,2]. The etiology of AD is multifaceted and varies among individuals, with significant correlations observed with genetic factors, acquired skin barrier dysfunction, immune abnormalities, and environmental factors [3]. The increasing incidence of AD has made it a global health concern, with existing treatment options, such as the antibody–drug interleukin, struggling to meet clinical demands. A 2020 report revealed that over 75% of physicians were dissatisfied with current treatments, with itch reduction being a pressing need for 75.8% of moderate-to-severe AD patients.

Atopic dermatitis (AD) is primarily characterized by an excessive immune response to exogenous substances such as food, animal dander, dust mites, and pollen, resulting in chronic eczema and itching. Continued scratching further damages the skin barrier, making it more susceptible to infection and subsequent tissue damage, thereby disrupting skin homeostasis [4]. Following skin barrier disruption, macrophages infiltrate and ingest allergens, triggering a range of inflammatory responses. This includes the release of cytokines and chemokines, activation of Th2 cells, and subsequent release of interleukin-4 (IL-4) and interleukin-13 (IL-13). These interleukins activate B cells to produce IgE, leading to mast cell and eosinophil activation and the release of more inflammatory cytokines and histamine, causing ad-related symptoms [5,6]. Studies have shown that eosinophils accumulate in skin lesions and produce inflammatory cytokines, including IL-4, which exacerbate atopic dermatitis [7]. In addition, allergen exposure to dermal dendritic cells (DCs) promotes IgE class switching, inducing a classic type 2 response [8]. The inflammatory response triggers a feed-forward loop and an itch–scratch cycle, further perpetuating the pathophysiologic mechanisms of AD. The sensitization of peripheral sensory nerve endings and higher cortical processing further induce itching and exacerbate scratch-induced mechanical trauma [9]. Human skin is rich in peptidergic and nonpeptidergic cutaneous sensory neurons that play a crucial role in these responses. The former secrete immunogenic peptides that drive DC migration and TH2 polarization, while the latter responds to pruritic immune stimuli and drives scratching behavior. The disruption of the balance between proinflammatory cytokine activity and barrier elasticity is a key factor in the chronicity of AD lesions and recurrent microbial infections [10].

Filament polyprotein (FLG) is a histidine-rich structural protein expressed in the granular layer of the stratum corneum that binds to keratin fibers in epithelial cells to form aggregates that play an important role in the formation and maintenance of the skin barrier [11]. Deficiencies in FLG can have a significant impact on the epidermal barrier, affecting both the organization of cytoskeletal keratin filaments and the integrity of the keratinized envelope (CE) [11,12]. Protein hydrolytic processing promotes the aggregation of its monomers with cytoskeletal keratins within keratinocytes, contributing to the formation of flattened squamous structures of individual cells [13]. As keratinocyte differentiation proceeds, FLG monomers are further degraded by various hydrolytic enzymes into amino acids and their derivatives with water-absorbing functions, which together constitute the natural moisturizing factors of the epidermis, helping to maintain epidermal hydration and play a key immunomodulatory role [14]. FLG is histidine rich, and its metabolites help to maintain the acidic pH of the epidermis, playing an important role in inhibiting the metabolism of bacterial ceramides and regulating epidermal differentiation [15,16]. A decrease in FLG degradation products leads to an increase in the pH of the stratum corneum and increases specific protease activity, which promotes the release of proinflammatory mediators from the stratum corneum and induces an inflammatory response [17]. The precursor of FLG, profilaggrin, is encoded by the FLG gene on chromosome 1q21, and this loss-of-function (LOF) mutation is a major genetic risk factor for the development and persistence of atopic dermatitis [18]. A recent whole-genome sequencing study emphasized the association between variants within FLG, FLG2, HRNR, and TCHH1 and higher atopic dermatitis severity scores [19]. Experimental data showed reduced expressions of these genes following in vitro and in vivo recombinant IL-4 and IL-13 stimulation, with more pronounced reductions in AD lesion skin [20,21].

The ultrastructure showed that FLG gene loss-of-function mutations were associated with the disturbed arrangement of keratin fibers in the cytoskeleton and structural abnormalities in the keratin envelope, and these alterations led to the disruption of the epidermal barrier, which in turn increased the chances of allergen exposure [7]. Mutations in the function of the FLG gene also resulted in decreased levels of natural moisturizing factors, weakened SC hydration, and increased trans-epidermal water loss, which manifested itself as skin dryness [22]. Filaggrin plays an important role in protecting the skin barrier, retaining skin moisture, reducing inflammatory responses, and maintaining skin health, and it has a positive effect on the treatment of atopic dermatitis [3].

rhFLA-10 is an active functional protein independently developed by the team: its structure contains recombinant filaggrin functional fragments; the experimental design of the synthesis of a series of proteins was screened for pharmacological activity after the experimental screening of the rhFLA-10 on atopic dermatitis; and the repair of damage to the skin cell has good pharmacological activity. Therefore, the supplementation of rhFLA-10 may become a new research idea for the treatment of atopic dermatitis.

As previously stated, filaggrin, a key protein in the epidermis, plays a pivotal role in the formation and maintenance of skin barrier function. Current research has discovered that mutations in the filaggrin gene are associated with skin-related diseases such as atopic dermatitis and psoriasis. Consequently, our team has independently developed a novel recombinant human filaggrin (rhFLA-10) that contains active functional fragments of recombinant filamentous protein in its structure and plays a critical role in repairing skin barrier function. Subsequently, a series of extracellular matrix proteins were synthesized through a pre-experimental design [23,24,25,26], and, after experimental screening, it was demonstrated that rhFLA-10 possesses pharmacological activity against atopic dermatitis, particularly in the repair of damaged skin cells. Therefore, we believe that rhFLA-10 is a promising candidate drug for the treatment of atopic dermatitis, and we have selected it for further in-depth in vitro and in vivo therapeutic studies using atopic dermatitis models.

## 2. Materials and Methods

### 2.1. Construction of Recombinant Human Filaggrin (rhFLA-10) Expression Vector

The codon of the gene encoding the human filaggrin was adapted to *Escherichia coli* (*E. coli*) preferences with the addition of BamH1 and Nde1 double-enzymatic cleavage sites, and synthesized by Kingsley Biotechnology Co., Ltd (Nanjing, China). The DNA fragment was then used as a template and amplified according to the amplification system using polymerase chain reactions in the latter experiments.

The sequences of rhFLA-10 and the empty vector pET20b (Invitrogen, Guangzhou, China) were digested with BamH1 and Nde1 endonuclease (Takara Bio, Tianjin, China), respectively, both of which were reacted in a water bath for 1 h at 37 °C. After the rhFLA-10 DNA fragments and the pET20b empty vector were recovered and verified using nucleic acid electrophoresis gel, the rhFLA-10 gene sequence fragments were ligated to the digested products of pET20b using T4 DNA ligase (Takara Bio, Tianjin, China) and placed in a water bath at 16 °C for 3 h and subsequently transferred into *E.coli* DH5α (Invitrogen, Guangzhou, China, ATCC^®^ BAA-3219™). Following polymerase chain reaction amplification and identification, the above ligated products were transformed into *E. coli* BL21(DE3) (Invitrogen, Guangzhou, China, ATCC^®^ BAA-1025^TM^), and the recombinant plasmids were obtained by screening for positive colonies named *E. coli* BL21(DE3)/pET20b-rhFLA-10 [27].

### 2.2. Expression and Identification of rhFLA-10

Positive monoclones were inoculated in luria broth (LB, Amp^+^) medium for an overnight culture at 37 °C. The culture was then transferred (1%) to fresh LB medium (5 mL) and further incubated at 37 °C. The culture was grown to OD_600_ = 0.6 and induced, respectively, using different concentrations (0, 0.3, 0.6, 0.9 mg/mL) of IPTG at 37 °C for 1, 2, 3, and 4.5 h. Subsequently, the culture was collected, and the expression of rhFLA-10 was identified through SDS-PAGE analysis and Western blot analysis.

### 2.3. Purification and Purity Identification of rhFLA-10

The cells were harvested via centrifugation and resuspended in a bactericidal solution (phosphate buffer (20 mM; pH 6)). Then, the cells were lysed using a high-pressure homogenizer (KS-2000M, KAI BAI SI Nanotechnology (Shanghai) Co., Ltd., Shanghai, China), once at atmospheric pressure and twice at 800 Pa of pressure at 4 °C. The whole cell lysate was collected and centrifuged at 20,000 rpm for 20 min at 4 °C to obtain the supernatant samples. The solution at various stages of the nickel column purification process [28] was collected, and the purity was checked using SDS-PAGE and high-performance liquid chromatography (HPLC).

### 2.4. Effect of rhFLA-10 on HaCaT Cell Viability

HaCaT cells (Human Immortalised Keratinocytes, Chinese Academy of Sciences, Shanghai, China, ATCC^®^ no. BNCC101683) were cultivated in Dulbecco modified eagle medium (high glucose, Gibco, Carlsbad, CA, USA) (H-DMEM) enhanced with 10% fetal bovine serum (FBS) in a humidified atmosphere at 37 °C with 5% CO_2_.

HaCaT cells (4000–6000 cells/well) in the logarithmic growth phase were inoculated into 96-well cell culture plates for 24 h. After that, the cells were treated for 24 and 48 h, as follows: the control group was administered H-DMEM with 1% FBS (100 μL per well), and the rhFLA-10 group was administered H-DMEM with 1% FBS using different rhFLA-10 concentrations (5, 10, 20, 40 μg/mL) (100 μL per well). Finally, the effect of rhFLA-10 on cell viability was detected using an MTT assay according to the manufacturer’s instructions. The cell viability was determined using the following equation: cell viability = OD value of experimental group/OD value of control group × 100%.

### 2.5. β-Hexosaminidase Assay

Mouse mastocytoma P815 cells (ATCC^®^ TIB-64™) were purchased from the Chinese Academy of Sciences (Shanghai, China) and cultured in H-DMEM with 10% FBS and 1% penicillin/streptomycin (100 units/mL) in a humidified atmosphere at 37 °C with 5% CO_2_.

The cells (5 × 10^5^ cells/well) were seeded into 24-well cell culture plates for 24 h. Then, the cells were induced with C48/80 and drugs for 1 h to activate degranulation as follows: the control group was not treated, the model group received H-DMEM with 20 μg/mL C48/80, the Dipotassium glycyrrhizinate group (DG, positive control group) received H-DMEM with 20 μg/mL C48/80 and 100 μg/mL DG, and the rhFLA-10 group received H-DMEM with 20 μg/mL C48/80 and 5, 10, and 20 μg/mL rhFLA-10.

Afterward, 50 μL of cell supernatant from each well was transferred into a 96-well culture plate to examine the cell degranulation rate according to the manufacturer’s instructions. The release rate of β-hexosaminidase was determined as the degranulation rate of cells in various treatment groups. The formula is as follows: degranulation rate = absorbance of supernatant/(absorbance of supernatant + absorbance of lysate) × 100%.

### 2.6. Neutral Red Staining Assay

As in the 2.5 step, the cells were collected and then treated according to the manufacturer’s instructions. After that, the morphology and quantity of the cells were observed under a microscope.

### 2.7. Expression of M1-Type Macrophage Markers

Primary precursor cells were obtained and cultured from the bone marrow of male C57BL/6 mice (6–8 weeks, 20–25 g). The cells were then stimulated to differentiate into monocyte-derived macrophages with 25 ng/mL macrophage colony-stimulating factor (M-CSF1) and cultured in DMEM with 10%FBS for 7 days. Then, differentiated macrophages were exposed to inducers of M1 macrophages (100 ng/mL LPS and 20 ng/mL IFN-γ) for 24 h, followed by stimulation with rhFLA-10 (2.75, 5.5 μg/mL) for an additional 24 h. Total RNA was extracted from cells using a HiPure Fibrous RNA Plus Kit (Magen, Guangzhou, China) and was subsequently transcribed into cDNA using a reverse transcription kit (Tiangen Biotech, Beijing, China). qRT–PCR was subsequently performed with a SYBR-Green Quantitative PCR Kit (Bio-Rad, Hercules, CA, USA) and the CFX96 Touch Real-Time PCR Detection System (Bio-Rad, Hercules, CA, USA). The relative expressions of IL-1β, TNF-α, IL-10, and Arg-1 were quantified using the 2^−ΔΔCt^ method, with GAPDH serving as the reference gene [29].

### 2.8. Uptake of rhFLA-10 by HaCaT

Protein staining was performed as follows: dissolve Cy5 NHS Ester in anhydrous DMSO or DMF to make a stock solution with the concentration of 10 mg/mL. add the Cy5 NHS Ester solution to the rhFLA-10 solution, making the starting point a 3:1 molar ratio of dye to protein; and incubate the reaction at room temperature for 2 h in the dark.

HaCaT cells (2 × 10^4^ cells/well) were inoculated into 24-well plates with cell crawlers placed. When the cell fusion degree reached over 90%, the cells were treated with rhFLA-10 (20 μg/mL). Then, the cell plates were harvested at 0, 2, 4, and 8 h, respectively, and washed three times with PBS, and fixed with 4% paraformaldehyde solution at 4 °C for 10 min. After that, the cell plates were washed three times with PBS to remove unfixed cells and residual 4% paraformaldehyde solution. A total of 100 μL of phalloidin was added to each well, incubated for 15–20 min, washed three times with PBS, and then stained and blocked on the plate with DAPI staining after a little drying. The fluorescence intensity of cells in each group was observed via confocal microscopy [30]. The expected results of the experiment were that the rhFLA-10 would be stained red with Cy5 dye, for localization; the cytoskeleton, stained green with FITC dye; and the nucleus, stained blue with DAPI.

### 2.9. Percutaneous Penetration of rhFLA-10 in C57BL/6 Mice

Animals: C57BL/6 mice were purchased from Guangdong Animal Center (No. 44007200069979) and kept in an appropriately controlled environment. The experimental protocol used in this study was approved by the Institutional Animal Care and Use Committee of Jinan University (approval number 20220302-18). All experiments were conducted in accordance with the Chinese Guidelines for the Care and Use of Animals and were approved by the Animal Ethics Committee of the Chinese Academy of Medical Sciences.

The skin of C57BL/6 mice was depilated one day in advance with a shaving razor and Viton depilatory cream and sterilized the next day. Two areas were delineated in the abdominal depilation area of each mouse, one as the blank group and the other as the rhFLA-10 administration group. A total of 200 μL of rhFLA-10 (20 μg/mL) was evenly applied to the corresponding area of the back skin, while the blank group was left untreated. Mice were humanely executed at 2, 4, and 8 h after administration, and the skin tissue at the administration site was excised, neatly embedded in OCT, and snap-frozen in liquid nitrogen. Afterward, frozen sections were created, with a section thickness of 10 μm, fixed through immersion in acetone for 10 min, and air-dried. The sections were observed using a fluorescence microscope and photographed.

### 2.10. Induction of Atopic Dermatitis Model and Treatment with rhFLA-10 Gel

Animals: Thirty KM mice were purchased from Guangdong Animal Center (No.: 44007200069979) and kept in an animal laboratory under the standards of the specific pathogen (SPF). The series of animal experimental schemes in this study conformed to the IACUC guidelines and were approved by the Experimental Animal Ethics Committee of Jinan University.

Model preparation and drug administration in groups: 30 mice were randomly allocated into 5 groups with 6 mice in each: a Normal group (untreated on the back during the whole process), Vehicle group (using 125 mg IMQ to establish atopic dermatitis model + blank poloxamer for back application), Paeonol group (positive control group, 125 mg IMQ + 200 μL Paeonol gel (containing 3 mg/g Paeonol) for back application), Low-rhFLA-10 group (125 mg IMQ + 200 μL rhFLA-10 gel (0.2 g/g) for back application), and High-rhFLA-10 group (125 mg IMQ + 400 μL rhFLA-10 protein gel (0.2 g/g) was applied to the back). The mice were anesthetized with 2% pentobarbital sodium (80 mg/kg) and then shaved and depilated on their backs using a razor and Viton, creating a 3 cm × 3 cm model area. The day of depilatory treatment was designated as day 0, and the mice were allowed to rest for one day before administration. Except for the normal group, 125 mg of imiquimod cream was applied to the back modeling areas of the mice in the other groups starting from Day1 to sensitize the mice, and we continued to apply IMQ cream every day to stimulate and maintain the inflammatory reaction on the back modeling areas for 7 days [31]. The skin condition of the mice was then monitored and recorded via photography.

### 2.11. Mouse Scratching Behavior Assay

After 30 min of each drug application, the mice were placed in an observation cage alone for us to observe the number of times that the mice scratched the modeling areas within 10 min, with consecutive scratching instances considered a single event [32]. After the last stimulation (day 7), we observed and recorded the number of times that the mice scratched the modeling areas within 10 min to evaluate the scratching behavior. Then, we anesthetized the mice, observed the treatment of back dermatitis, and collected skin tissue samples.

### 2.12. Detection of the Extent of Skin Lesions in Mice

After anesthetizing the mice with 2% pentobarbital sodium anesthetic via intraperitoneal injection (80 mg/kg), the skin damage on the back of the mice was observed and recorded every day to evaluate the severity of inflammation in the skin damage of the mice. This evaluation was based on five distinct symptoms: dryness, eyebrow loss, hemorrhage/rash, ulcerated epidermal detachment, and edema. Each symptom was assigned a score ranging from 0 (none) to 3 (severe), with the final score being the sum of these individual symptom scores (Table 1). The overall skin tissue inflammation was then rated on a scale of 0–12 [33].

### 2.13. Histopathologic Examination of the Skin

The back skin injury tissue was obtained after humanitarian execution at different time points under aseptic conditions and fixed in a 10% formaldehyde solution. Standard procedures including routine paraffin embedding, sectioning, HE and Masson staining, and the microscopic observation of histopathological changes were performed. Additionally, molecular indicators such as filipin were tested using immunohistochemistry. Using H&E-stained tissue sections, the epidermal thickness of each group was measured using Image Pro Plus software (version 6.0). Using Masson-stained tissue sections, the collagen deposition rate of each group was quantified using ImageJ software (version 1.8.0) [34].

### 2.14. Statistical Analysis

All data were expressed as the mean (mean) ± standard deviation (SD) of at least three independent experiments. Statistical analyses were performed using GraphPad Prism 9 software (GraphPad Software Inc., Jolla, CA, USA). Differences between more than two groups were analyzed using one-way ANOVA followed by Tukey’s HSD comparison test. Statistical significance was set at *p* < 0.05.

## 3. Results

### 3.1. Expression and Purification of rhFLA-10

The recombinant plasmid pET20b-rhFLA-10, which contains two 6-histidine tags for facilitating protein purification [35], was generated through enzymatic ligation (Figure 1A). Monoclonal 6 was confirmed as a positive recombinant via agarose gel electrophoresis and PCR analysis, exhibiting bands of identical size to those of plasmid rhFLA-10 (approximately 1400 bp). This observation suggests the successful transformation of plasmid rhFLA-10 (Figure 1B). The expressed rhFLA-10, with a molecular weight of 55 kD, was identified through SDS-PAGE and Western blot analysis under the conditions of 37 °C, a supplementation of 0.5 mg/mL IPTG, and being cultured for 4 h (Figure 1C). The results demonstrate that in the absence of added IPTG, the protein expression shows a significant upward trend over time, peaking at 3 h and 4.5 h post induction. In contrast, an increase in the IPTG concentration resulted in a decrease in the rhFLA-10 expression. This suggests that the rhFLA-10 gene exhibits leaky expression, enabling the maintenance of cellular physiological functions even without an inducer, thereby achieving the highest expression level of the target protein [36]. Consequently, the optimal conditions for the expression of the recombinant protein A-10 were determined to be at 3 h and 4.5 h post induction, without the addition of IPTG (Figure 1D). The purity of rhFLA-10 was 96.75% as determined via HPLC (Figure 1E).

### 3.2. rhFLA-10 Inhibited the Degranulation of Mast Cells

The cytotoxicity evaluation of rhFLA-10 using the MTT assay indicated that the cell state was not significantly different from the control group after 24 to 48 h of treatment with rhFLA-10 at a concentration of 5–20 μg/mL, and the cell outline was clear and the vitality was good, indicating that rhFLA-10 would not have any toxic effect on HaCaT cells at a concentration range of 5–20 μg/mL and that the cells can grow normally (Figure 2A).

To evaluate the effect of rhFLA-10 on the degranulation of mast cells, P815 cells were treated with rhFLA-10 at different concentrations of 5, 10, and 20 μg/mL in the treatment group. The results show that the proportion of degranulated cells in the Model group was significantly higher than that in the Control group (*p* < 0.001), and the cells were mostly opaque, with inflated and irregular shapes, and the cytoplasm was stained red, which indicates a typical degranulation state. The degree of cell degranulation in the positive control (100 μg/mL DG) was lighter than that in the Model group, which was close to the cell state of the Control group, which indicates that the inducer was effective on cells and that the cell model was successfully prepared, proving the validity of this experiment. With the increase in the rhFLA-10 concentration, the degranulation status of mast cells excited by C48/80 was gradually improved. When the protein concentration increased to 20 μg/mL, the cell state tended to be normalized, and the degranulation rate of cells was significantly lower than that of the model group (Figure 2C). Semi-quantitative results once again proved that the concentration of rhFLA-10 at 20 μg/mL had a significant effect on mast cell degranulation (*p* < 0.01). The above results show that different concentrations of rhFLA-10 have obvious inhibitory effects on mast cell degranulation and could play a soothing effect and that the inhibitory effect is more obvious at the dose of 20 μg/mL (Figure 2D).

As a common glycoside hydrolase, β-aminohexosidase widely exists in the biological world, and it is a sign that mast cells release particles [37]. The semi-quantitative results of β-aminohexosidase release show that the treatment of the rhFLA-10 group embodied a certain therapeutic effect. The release of β-hexosaminidase in the model group was significantly increased, and the release of β-hexosaminidase in the positive control group was significantly reduced compared with the control group (*p* < 0.001), which once again indicates that the cell model was successfully established. Compared with the Model group, the rhFLA-10 group with different concentrations can reduce the release of β-hexosaminidase, and the therapeutic effect was more pronounced in the rhFLA-10 group when the concentration was 20 μg/mL (*p* < 0.001), which shows that the rhFLA-10 protein can inhibit the degranulation of mast cells by reducing the level of β-hexosaminidase (Figure 2E).

### 3.3. Effect of rhFLA-10 on Phenotypic Differentiation of m1-Type Macrophages

Therefore, this study further detected the expression levels of related inflammatory factors in macrophages treated with rhFLA-10 protein. The results show that the expression levels of TNF-α and IL-1β in M1 macrophages decreased significantly with 100 nM rhFLA-10 protein at 24 h (*p* < 0.05), while the expression levels of Arg-1 and IL-10 increased significantly with the increase in the rhFLA-10 concentration (*p* < 0.05), indicating that the rhFLA-10 protein can inhibit the release of proinflammatory factors to inhibit the polarization of macrophages to M1 cells. rhFLA-10 promotes the polarization of M1-type cells to M2-type cells by up-regulating the expressions of Arg-1 and IL-10, thus improving the inflammatory symptoms (Figure 2B).

### 3.4. Uptake Behavior of rhFLA-10 in HaCaT Cells

The rhFLA-10 was stained red with Cy5 dye, for localization; the cytoskeleton was stained green with FITC dye; and the nucleus was stained blue with DAPI. At 0 h, it could be seen that the cells were closely attached to the bottom of the orifice plate in good conditions: a clear outline, moderate density, easy to observe, and good cell state. At 2 h, a small amount of rhFLA-10 was localized in the cell, and the fluorescence was gradually enhanced after 4 h. At 8 h, the fluorescence in the cell was brighter, but the fluorescence density was not significantly enhanced compared with that at 4 h, suggesting that the rate of entry of rhFLA-10 into the cell was gradually slowed down and that the quantity tended to be equilibrated starting at 4 h. These results indicate that rhFLA-10 can be rapidly absorbed by Hacat cells and enter the cell, thus exerting its function (Figure 3A).

### 3.5. Transdermal Absorption Properties of rhFLA-10 in Mice

According to the previous experimental results, rhFLA-10 can be absorbed by epidermal cells. Here, we found that rhFLA-10 can effectively penetrate into the stratum corneum of the skin, and some proteins can even penetrate into the dermis, thereby establishing the first line of defense between the body and the external environment. After rhFLA-10 was smeared on the skin for 2 h, the fluorescence was weakly distributed in the epidermis, and more in the pores, indicating that the drug had begun to penetrate the skin within 2 h. The fluorescence intensity of the epidermis gradually increased from 4 h to 8 h, and reached the highest value at 8 h, indicating that a large amount of rhFLA-10 had been absorbed by the skin, and a small amount of fluorescence could be detected in the dermis at 8 h. The results show that rhFLA-10 has certain transdermal absorption characteristics and can penetrate the skin layer (Figure 3B).

### 3.6. rhFLA-10 Treatment Study of Atopic Dermatitis in KM Mice

To evaluate the therapeutic effect of A10 on atopic dermatitis in mice, rhFLA-10 was administered to atopic dermatitis mice every day for one week, and the skin damage condition and itching times of mice were evaluated. Compared with the Vehicle group, the Paeonol group and the rhFLA-10 group both showed therapeutic effects on the 4th day, and the rhFLA-10 group showed more obvious effects (*p* < 0.01), and the skin damage condition did not aggravate significantly. At the end of day 7, the skin injury symptom score of the Vehicle group was the highest, and the back skin of the mice was rough and covered with thick yellow scales; the skin of the mice was obviously thickened and inelastic, and the mice were in poor condition with abnormal agitation. The Paeonol group still had obvious yellow scales, and the score of the skin injury symptoms was second only to that of the Vehicle group, while the rhFLA-10 group showed a better therapeutic effect with no obvious skin thickening or yellow scales. The mice were in good condition and the high-dose group was almost completely cured on day 7 (Figure 4A,B).

The therapeutic effect of rhFLA-10 was verified again through the behavioral observation of the mice. The number of times that the mice scratched their backs was recorded each day, which reflected the degree of agitated skin itching in the mice that day. The data of scratching times recorded on the last day show that the scratching times of mice in the Vehicle group were significantly increased compared with those of the Normal group (*p* < 0.001), and the scratching times of mice in the Paeonol group were significant, indicating that dermatitis symptoms were not alleviated. Compared with other groups, the scratching frequency of the rhFLA-10 group was less. The scratching frequency of the high-dose rhFLA-10 group was close to that of the Normal group (*p* < 0.001), which indicates that rhFLA-10 had better pharmacological activity in the in vivo treatment model of atopic dermatitis (Figure 4C).

### 3.7. Histopathologic Section Study of Skin Lesions in Mice

HE staining and Masson staining revealed the pathological changes in the mice’s back skin. In observing the pathological sections of mice’s skin, the skin epidermis of mice in the Vehicle group underwent significant thickening, the arrangement of collagen fibers was misarranged, and there was a large amount of inflammatory infiltration in the skin cells. After treating the skin with different concentrations of rhFLA-10, the above symptoms were improved. Compared with the Normal group, the low-rhFLA-10 group showed collagen that was more regularly arranged, and the epidermal integrity was better (*p* < 0.01); at the same time, the epidermal thickness in the high-rhFLA-10 group was almost restored to the normal level (*p* < 0.001), which indicates that the rhFLA-10 group has a good effect on the treatment of atopic dermatitis in in vivo conditions and can inhibit epidermal hyperplasia (Figure 5).

## 4. Discussion

Currently, the clinical symptoms of patients with atopic dermatitis have a serious impact on their quality of life, which is made even more problematic by the limited availability of clinical treatments [38]. Filaggrin, as a structural protein, has an important impact on the epidermal barrier. It enables the regular aggregation of keratin fibers, which leads to the contraction of the keratin-forming cytoskeleton and flattening of cells in the basal layer of the epidermis [39]. Meanwhile, filaggrin monomers further form natural moisturizing factors in the presence of various hydrolytic enzymes and other enzymes, which help maintain the hydration function of the epidermis. In this study, through an in-depth study of the structure and function of filaggrin, a new treatment option for atopic dermatitis was proposed, i.e., the use of recombinant human filaggrin (rhFLA-10), which contains a functional fragment of filaggrin. Recombinant plasmids, which are successfully constructed recombinant proteins, were expressed in this study, and high-quality and efficient recombinant human filaggrin (rhFLA-10) was synthesized through genetic engineering technology, with optimized expression conditions and isolation and purification, and high-quality and efficient rhFLA-10 were obtained. The optimal expression strain and induction temperature conditions for rhFLA-10 have not yet been screened experimentally, and more in-depth screening research will be conducted on this issue in the future. At the same time, the inherent reasons for the leakage expression of this protein will also be further explored through experiments, and the best optimization expression scheme will be search for [40].

Individuals with atopic dermatitis often suffer from chronic eczema accompanied by itching, which leads to repetitive scratching and recurrent damage to the skin barrier, rendering it more susceptible to microbial pathogen infections and inflammation caused by allergens, toxins, and irritants [4]. In this study, in vivo experiments utilizing an atopic dermatitis mouse model demonstrated that rhFLA-10 can reduce scale coverage, epidermal thickening, and erythema, thereby maintaining the stability of mouse skin. This evidence suggests that rhFLA-10 can effectively repair the skin barrier of atopic dermatitis and maintain the immune stability of the epidermis. Furthermore, it can significantly decrease the frequency of scratching in model mice, thereby avoiding the vicious cycle of repeated scratching leading to exacerbated skin damage and enhancing the therapeutic effect on atopic dermatitis (Figure 4), suggesting that rhFLA-10 has a potent pharmacological activity in an in vivo therapeutic model of atopic dermatitis, and has a favorable effect on the repair of atopic dermatitis.

AD is a chronic, recurrent inflammatory response characterized by hyperkeratosis, epidermal hyperplasia, and the accumulation of lymphocytes and mast cells [41]. Therefore, this study investigated the effects of rhFLA-10 on epidermal thickness and mast cell infiltration in AD mice. The results of the staining of mouse skin sections illustrated that rhFLA-10 treatment significantly alleviated the degree of epidermal hyperplasia and suppressed the inflammatory infiltrate (Figure 5). This finding is consistent with previous studies on the use of myricetin [42] and *Portulaca oleracea* L. [43] for the treatment of AD. However, the mechanism through which it exerts its specific therapeutic effects within the epidermal cells has not been clarified, and further studies are needed.

AD is an immunoallergic disease with a wide range of clinical implications. Progress is being made in understanding the development of AD, but its complex pathogenesis remains poorly understood, and no cure is currently available. Although rhFLA-10 has shown promising therapeutic effects in atopic dermatitis, it is still deficient. In this study, we constructed an acute atopic mouse model that does not exactly match the clinical atopic dermatitis phenotype. In the future, we hope to construct a mouse model that can match the human atopic dermatitis even better and measure the amount of plasma cytokines in mice during the treatment process in order to investigate the therapeutic mechanism of rhFLA-10 in the immunization process of the body.

The goal of this study was to address the difficult-to-treat nature of atopic dermatitis and to provide patients with a new therapeutic option to alleviate their symptoms and improve their quality of life. By gaining insight into the therapeutic mechanisms and effects of synthetic proteins, we expect to provide important information for the development of innovative treatments in the future. The results of this study suggest that rhFLA-10 may be a promising drug for the treatment of AD. However, further studies are needed to fully understand the molecular mechanisms of rhFLA-10 therapy and to more closely match the immunodeficient mouse model of clinical manifestations, so as to assess its potential in clinical applications. In addition, future studies could explore the possibility of combining rhFLA-10 with other therapeutic agents to enhance its efficacy and minimize potential side effects.

## Figures and Tables

**Figure 1 bioengineering-11-00426-f001:**
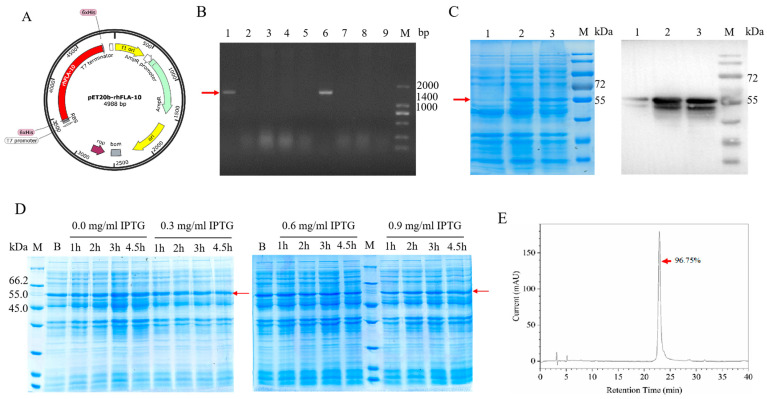
Construction, expression, and identification of rhFLA-10. (**A**) Recombinant pET20b-rhFLA-10 plasmid profile. (**B**) Agarose gel electrophoresis for colony PCR identification. Lane 1: recombinant pET20b-rhFLA-10 plasmid; Lanes 2~9: picked monoclonal cells; M: DL2000 DNA marker. (**C**) SDS-PAGE and Western blot identification of rhFLA-10 expression. Lane 1: rhFLA-10 expression before induction; Lanes 2 and 3: rhFLA-10 expression after induction; M: protein molecular weight marker. (**D**) Induction expression and SDS-PAGE of rhFLA-10 under different concentrations of IPTG and time conditions. B: rhFLA-10 expression before induction; M: protein molecular weight marker. (**E**) HPLC analysis of protein purity of rhFLA-10.

**Figure 2 bioengineering-11-00426-f002:**
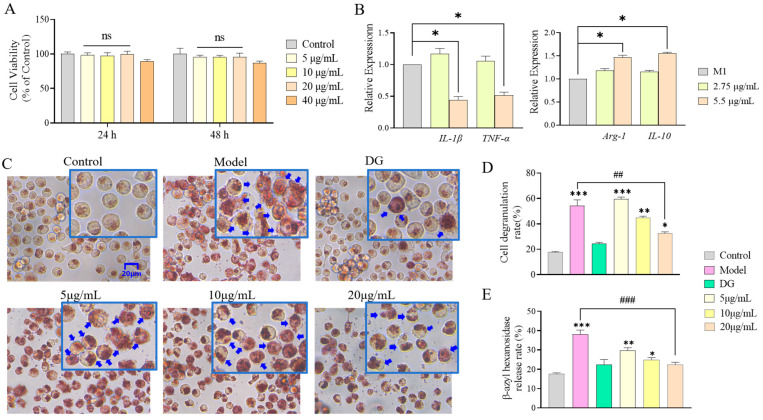
Effects of rhFLA-10 on the physiological state of cells. (**A**) MTT assay used to detect the effect of rhFLA-10 on the activity of HaCaT cells at 24 h and 48 h. Note: ns = No Significant. (**B**) Expression of inflammatory markers TNF-α, IL-1β, Arg-1, and IL-20 in M1 macrophage treated with rhFLA-10 24 h. (**C**) Detection of the effect of rhFLA-10 on P815 cell degranulation at different doses. DG: Dipotassium glycyrrhizinate. Note: The blue boxes are localized enlargements of each figure, and the blue arrows indicate degranulated cells. (**D**) Semi-quantification of cell degranulation rate in P815 cells at different doses. (**E**) Semi-quantification of the release of β-aminohexokinase in P815 cells at different doses. (*n* = 3, Mean ± SD, * *p* < 0.05; ** *p* < 0.01; *** *p* < 0.001 vs. Control group; ## *p* < 0.01; ### *p* < 0.001 vs. Model group).

**Figure 3 bioengineering-11-00426-f003:**
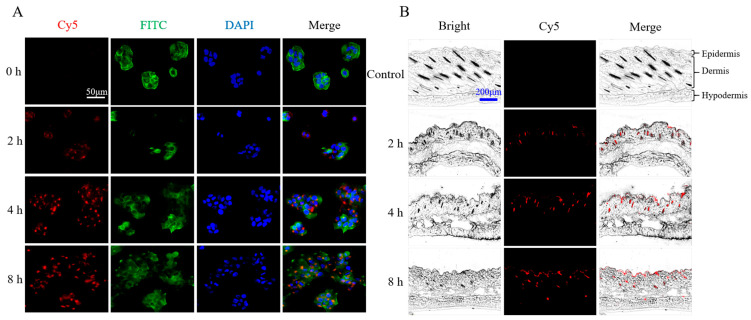
Immunofluorescence localization of rhFLA-10 in HaCaT cells and the skin of C57BL/6 mice. (**A**) The timing process of rhFLA-10 uptake by HaCaT cells was observed via immunofluorescence colocalization (scale bar = 50 μm). Note: The red represents rhFLA-10, the green represents the cell skeleton, and the blue represents the cell nucleus. (**B**) Transit time course of rhFLA-10 into the skin of C57BL/6 mice (scale bar = 200 μm). Note: The three layers of skin structure from top to bottom of the slice are the epidermis, dermis, and hypodermis.

**Figure 4 bioengineering-11-00426-f004:**
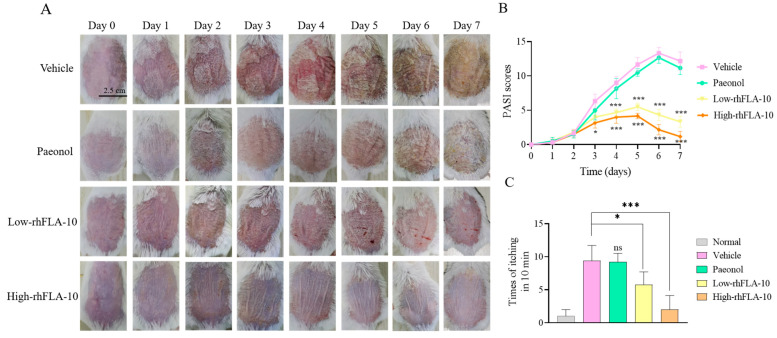
Therapeutic evaluation of rhFLA-10 on atopic dermatitis model mice. (**A**) Daily skin condition of mice in each group (scale bar = 2.50 cm). (**B**) Statistics of daily atopic dermatitis skin damage PASI score of mice. (**C**) Scratch times of mice in each group on day 7. (*n* = 6, Mean ± SD, ns = No Significant; * *p* < 0.05; *** *p* < 0.001 vs. Vehicle group).

**Figure 5 bioengineering-11-00426-f005:**
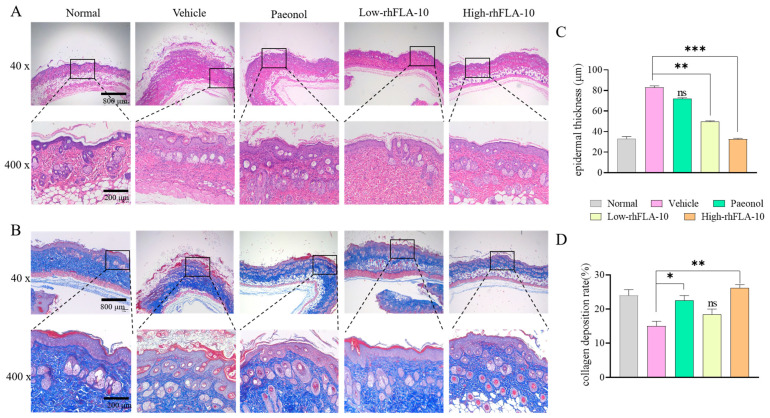
Histological observations of skin lesions in mice. (**A**) H&E pathology section of mice skin treated with rhFLA-10 on day 7 after treatment (40×: scale bar = 800 μm; 200×: scale bar = 200 μm). (**B**) Masson pathology section of mice skin treated with rhFLA-10 on day 7 after rhFLA-10 expression before induction (40×: scale bar = 800 μm; 200×: scale bar = 200 μm). Note: The three layers of skin structure from top to bottom of the slice are the epidermis, dermis, and hypodermis. (**C**) Epidermal thickness of each group of mice on day 7 after treatment. (**D**) The collagen deposition rate of each group of mice on day 7 after treatment. (*n* = 3, Mean ± SD, ns = No Significant; * *p* < 0.05; ** *p* < 0.01; *** *p* < 0.001 vs. Vehicle group).

**Table 1 bioengineering-11-00426-t001:** Mouse skin PASI scoring standard.

Observation Item	Score	Score Standard
Skin thickness	0	Smooth skin without folds
	1	The skin appears slightly wrinkled
	2	The skin is all slightly wrinkled
	3	The skin folds further
	4	Severe wrinkling accompanied by weight loss or poor condition
Chippings covering condition	0	The skin is smooth without scales
	1	The skin appears slightly scaly
	2	Slightly scaly skin covered
	3	The skin is completely covered with scales and thickened
	4	Heavy chippings accompanied by weight loss or poor condition
Erythema dermis	0	The skin is smooth
	1	The skin has slight erythema
	2	The skin is completely covered with erythema
	3	The severity of erythema is further deepened
	4	Severe erythema with weight loss or poor condition

## Data Availability

The raw data supporting the conclusion of this article will be made available by the authors without undue reservations.

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
