# Peer review of "A Novel Recombinant Human Filaggrin Segment (rhFLA-10) Alleviated a Skin Lesion of Atopic Dermatitis"

_bioengineering, 2024, doi:10.3390/bioengineering11050426_

Round 1
Reviewer 1 Report
Comments and Suggestions for Authors
Jiawen Zhu and colleagues report in their original and informative article the construction of a new recombinant human filaggrin (rhFLA-10) bacterial plasmid expression vector. The expression product, an active form of human filaggrin, was then extensively tested on cell lines and in a mouse model for atopic dermatitis (AD). The findings support the authors’ conclusion of a potential new treatment option in the future for AD.
Although some descriptions of standard procedures could be shortened, the manuscript needs only minor editing. Figures and English are good.
Very minor issues detected:
Check and adjust/correct layout of all references, e.g., ref #1 “… Atopic dermatitis nat rev dis primers…” – Should be more like “… Atopic dermatitis. Nat Rev Dis Primers…”
L. 238: check “cell fusion degree reached” – suggestion: perhaps use “cell confluence reached” or “cell confluency reached”
Comments on the Quality of English LanguageEnglish is good, only very minor punctiation issues.
Reviewer 2 Report
Comments and Suggestions for Authors
The author of this article designed a new recombinant human silk fibroin (rhFLA-10) expression vector and synthesized and purified the protein. The author found through research that low concentrations of rhFLA-10 are non-toxic to cells and also have therapeutic and anti-inflammatory effects on AD. These findings are of great significance for the treatment of AD. I suggest that this article be published after minor revisions.
1, Most of the Figures are too small, and the font inside the pictures is also too small to see clearly.
2, Writing should be serious, for example, there is a editing mode on the first line of page nine.
3, Each Figure should be annotated clearly. For example, what do different colors in Figure 3 represent? Each structure of the skin should be labeled.
Author Response
请参阅附件。

Reviewer 3 Report
Comments and Suggestions for Authors
In this paper, the authors intend that the human filaggrin segments they designed will be effective in treating atopic dermatitis. However, since this paper is the first to publicize the medicinal efficacy of the human filaggrin segment designed by the authors, it is necessary to present the necessary evidence to help readers understand. As a reviewer, I will suggest the following specific improvements.
1) Describe in the Introduction how the decision was made to use rhFLA-10 for the treatment of atopic dermatitis, showing the authors' publications.
2) It is difficult to understand the effect of rhFLA-10 from the image in C of Figure 2, so please display it in an easy-to-understand manner.
3) There is an explanation that D in Figure 2 is a semi-quantitated image of C, but please add a specific method for quantifying from the image to the explanation.
4) The distribution of rhFLA-10 in the skin shown in the explanation of B in Figure 3 is difficult to understand, so please display it in an easy-to-understand manner.
5) The photos in Figure 4 A should be displayed in an easy-to-understand manner, as they are each small and difficult to compare.
6) Regarding the experimental results of the atopic dermatitis mouse model using KM mice, is this mouse an established method for human atopic dermatitis model? If you have any publications to that effect, please cite them. We also propose to show quantitative values of plasma cytokines in each group of mice as therapeutic effects.
7) Concerning the discussion, the experimental results shown in this paper should be discussed with a focus on the therapeutic effects of rhFLA-10.
Author Response
请参阅附件。
